# Artificial Intelligence Models and Tools for the Assessment of Drug–Herb Interactions

**DOI:** 10.3390/ph18030282

**Published:** 2025-02-20

**Authors:** Marios Spanakis, Eleftheria Tzamali, Georgios Tzedakis, Chryssalenia Koumpouzi, Matthew Pediaditis, Aristides Tsatsakis, Vangelis Sakkalis

**Affiliations:** 1Department of Toxicology and Forensic Sciences, School of Medicine, University of Crete, 71003 Heraklion, Greece; tsatsaka@uoc.gr; 2Computational Bio-Medicine Laboratory, Institute of Computer Science, Foundation for Research and Technology—Hellas, 70013 Heraklion, Greece; tzamali@ics.forth.gr (E.T.); gtzedaki@ics.forth.gr (G.T.); koumpouzi@ics.forth.gr (C.K.); mped@ics.forth.gr (M.P.); sakkalis@ics.forth.gr (V.S.)

**Keywords:** drug–herb interactions, herbal medicines, artificial intelligence, interactions, machine learning, deep learning, knowledge graphs, chemoinformatics, network pharmacology, dietary supplements, XAI

## Abstract

Artificial intelligence (AI) has emerged as a powerful tool in medical sciences that is revolutionizing various fields of drug research. AI algorithms can analyze large-scale biological data and identify molecular targets and pathways advancing pharmacological knowledge. An especially promising area is the assessment of drug interactions. The AI analysis of large datasets, such as drugs’ chemical structure, pharmacological properties, molecular pathways, and known interaction patterns, can provide mechanistic insights and identify potential associations by integrating all this complex information and returning potential risks associated with these interactions. In this context, an area where AI may prove valuable is in the assessment of the underlying mechanisms of drug interactions with natural products (i.e., herbs) that are used as dietary supplements. These products pose a challenging problem since they are complex mixtures of constituents with diverse and limited information regarding their pharmacological properties, especially their pharmacokinetic data. As the use of herbal products and supplements continues to grow, it becomes increasingly important to understand the potential interactions between them and conventional drugs and the associated adverse drug reactions. This review will discuss AI approaches and how they can be exploited in providing valuable mechanistic insights regarding the prediction of interactions between drugs and herbs, and their potential exploitation in experimental validation or clinical utilization.

## 1. Introduction

Herbal medicinal products, along with nutraceuticals, continue to grow in popularity, either as part of rational phytotherapy with well-established therapeutic applications or as integral components of traditional approaches aimed at promoting health and wellness. However, with the increasing use of both conventional drugs and herbal supplements, the potential for drug–herb interactions (DHIs) has become a critical issue in clinical practice [1,2,3]. Unlike drug–drug interactions (DDIs), which have been extensively studied, DHIs are less well understood due to the complex and often poorly characterized nature of herbal components. Nevertheless, DHIs can also result in reduced therapeutic efficacy, adverse effects, or even toxicities, making their effective prediction and management essential [4]. DHIs, like DDIs, arise when the pharmacological effects of a drug are altered by the presence of an herbal compound. The pharmacological mechanisms can be related to pharmacokinetic processes affecting drug absorption, distribution, metabolism, or excretion (PK-DHIs) or pharmacodynamic mechanisms altering the drug’s pharmacological effect on specific targets (PD-DHIs). PK-related interaction mechanisms primarily involve the inhibition or induction of drug transporters, such as P-glycoprotein (P-gp) and organic anion or cation transporters (OATPs, OCTPs), as well as metabolic enzymes of phase I metabolism, particularly those in the CYP450 family (e.g., CYP3A4/5, CYP2C9, CYP2D6), or phase II metabolism, like UDP-glucuronosyltransferases (UGTs) [5,6,7]. On the other hand, PD-related interaction mechanisms involve synergistic or antagonistic effects on a drug’s primary targets or secondary pathways, leading to either modulation of its primary pharmacological action or enhancement of side effects, thereby increasing the risk of adverse drug reactions (ADRs) [8,9,10].

Predicting DHIs is challenging due to the complexity of herbal products, which are characterized by their multicomponent nature, variable composition, and diverse biological activities [11]. Unlike single-entity pharmaceutical drugs, herbal products contain mixtures of bioactive compounds (phytochemicals) with potential multipotent actions, making them difficult to analyze using well-established experimental approaches validated for DDIs. While these research methods are essential, they often fall short in providing comprehensive and timely insights into DHIs because of the numerous variables involved [12]. The multiple active constituents in herbs, each with their own PK and PD profile, can vary even between batches of the same herb due to factors like plant origin, harvesting conditions, and processing methods, leading to inconsistencies in chemical profiles and biological effects. Furthermore, herbal products can interact with drugs simultaneously through both PK and PD mechanisms at primary and secondary targets. So, on one side, predicting DHIs solely based on individual components may leave out additional parameters, whereas the use of crude products in in vitro/in vivo studies often lacks reproducibility or even presents difficulties in extrapolating the results at clinical level [13,14,15]. A representative example is St. John’s Wort (SJW), a popular herbal remedy that attracted attention due to its multipotent action, especially in the central nervous system (CNS), and that eventually also became well known for its numerous potentially clinically significant DHIs [16,17,18]. SJW is a characteristic example of the multiparameter nature of DHIs. SJW contains several bioactive compounds like hypericin, hyperforin, and flavonoids. In the beginning of administration, hyperforin inhibits CYP enzymes (1A2, 2C9, 2C19, 2D6, and 3A4/5), potentially increasing drug exposure, while long-term use induces CYPs and P-gp, leading to reduced drug levels and increased metabolic clearance [19,20,21]. Additionally, SJW can elevate serotonin levels when combined with antidepressants, increasing the risk of serotonin syndrome [22]. Thus, it is important to find research approaches that can integrate multiparameter data for DHI prediction.

Computational or in silico pharmacology has revolutionized biomedical research and clinical practice by enabling the rapid analysis and prediction of drug behaviors through the exploitation of modeling and simulation (M&S) approaches [23]. In the context of DDIs, in silico pharmacology explores potential interactions by incorporating diverse data such as molecular structures, PK and/or PD parameters such as drug metabolism, transport, or binding on specific drug targets. Computational approaches for predicting DDIs can be broadly categorized into three main groups: (i) similarity-based methods; (ii) network-based methods; and (iii) machine learning (ML) methods [24]. Each category offers unique strengths and limitations, enabling different levels of insight into potential interactions based on the type and availability of drug data.

Similarity-based methods infer DDIs by evaluating similarity scores between drug profiles, such as structural, gene expression, and pharmaceutical profiles [25]. These methods are simple and interpretable, performing well when structurally similar drugs share common targets. However, they are susceptible to noise and thresholding issues, particularly when similarity metrics are prone to false positives, and they are limited when drugs lack structural similarity [26].

Network-based methods utilize networks like drug similarity networks or protein–protein interaction (PPI) networks to predict DDIs, leveraging connections between drugs through network inference [27]. Compared to direct similarity-based approaches, these methods are more robust against noise and can capture indirect interactions, such as drugs affecting the same pathway. However, PPI networks are still incomplete and contain noise, which can limit their accuracy. Additionally, indirect relationships may complicate the biological interpretability of predictions [28].

ML methods integrate diverse data sources to capture multiple aspects of drug data, including adverse reactions, target similarity, and signaling pathways [29]. Data integration enriches drug descriptions, often boosting predictive performance, and deep learning models handle complex, high-dimensional data effectively [30,31,32]. Nonetheless, the increased data complexity and dependency on complete datasets can hinder model performance when input data are missing. Furthermore, integrating numerous data sources may reduce interpretability, making it challenging to understand the molecular mechanisms underlying predictions. The introduction of ML and artificial intelligence (AI) tools has further enhanced this field by allowing for the processing of vast datasets, uncovering complex biological patterns, and predicting drug behaviors via data integration towards what is called translational pharmacology [24,33,34]. These advancements have accelerated drug discovery, reduced the need for extensive in vitro and in vivo testing, and even provided personalized medicine insights by integrating patient-specific data. As for DDIs, ML and AI have gained a lot of attention due to the often-efficient prediction of potential interactions by analyzing complex datasets [35,36,37]. AI approaches significantly advanced our understanding by uncovering multi-target interactions. In particular, AI models can integrate diverse data sources—including molecular structures, biological networks, and clinical data—making DDIs predictions more scalable, accurate, and insightful for understanding adverse effects and synergistic potentials [38,39,40,41]. These approaches are significantly reducing the time and cost associated with traditional experimental methods while increasing the accuracy and scale of interaction predictions. By identifying possible interactions early in the drug development process, in silico tools have enhanced drug safety, have guided DDIs experimental and clinical studies, and have contributed to better clinical decision making [30,32,42].

AI and ML can manage and integrate large and diverse datasets, such as cheminformatics data, pharmacological pathways, genomic data, and real-world evidence from clinical reports [29,43,44]. This integration allows for a more comprehensive analysis of potential interactions that can guide the experimental procedures, as well as clinical guidance [45]. For example, ML/AI algorithms can uncover relationships between herbal constituents and drug metabolism enzymes and predict interactions that might not be apparent through traditional analysis [35,46]. Moreover, ML/AI tools are scalable, which is particularly valuable in screening large libraries of drugs and herbal products and in prioritizing candidates for further experimental validation [47]. In addition, AI tools can integrate information regarding pharmacological pathways with chemoinformatic data and potentially provide mechanistic insights into DHIs [48]. In addition to pharmacological parameters, AI tools can also include patient clinical information, such as multimorbidity, pharmacogenomic data or even demographic characteristics, which align with the vision of precision medicine where treatments are adjusted to individual patient profiles and thus provide potential DHI information tailored to patient characteristics [49].

Drug interactions are a major contributor to adverse effects and their early detection is crucial to preventing potentially severe clinical consequences. Moreover, considering the self-treatment habits for herbal supplements, unidentified DHIs may lead to altered drug efficacy or unexpected toxicities, further complicating treatment outcomes. Despite the challenges and limitations regarding DHIs—which include (i) the lack of standardized data on herbal compounds; (ii) variability in herbal composition due to source, preparation, and administration; (iii) the limited availability of high-quality, labeled datasets for training AI models; (iv) multiple interactions and potential higher-order dynamics considering that herbs are mixtures of bioactive compounds—the advancements in AI can be a feasible approach for predicting DHIs. AI tools can reveal potential cases that, until further experimentally explored, can prompt healthcare providers to exercise caution when advising patients or even assisting patients themselves to adopt enhanced safety precautions. By leveraging AI technologies, a more comprehensive approach to DHI detection can be achieved, ultimately contributing to ADR prevention and improved therapeutic decision making. The complex nature of herbal products can be better approached using AI tools that are capable of analyzing complicated datasets that include different types of data [50]. For example, through AI tools for DHIs, chemoinformatic data can be leveraged along with PK and PD parameters and other pharmacological pathways or even clinical data. AI methods can analyze large datasets, identify patterns, and predict interactions that might not be immediately apparent through traditional methods (Figure 1). This review discusses (i) the application of AI in predicting DHIs, focusing on various machine learning (ML), deep learning (DL) and other relevant AI methods, as well as (ii) the integration of chemoinformatic data and the role of pharmacological pathways and the effect of available clinical data. Given the limited available literature on DHIs, we present several AI approaches (see also Appendix A) that have been applied in perspective DDI studies and propose how they could be adapted to support the case of DHIs by using known examples from the literature.

## 2. Exploitation of Chemoinformatics, Pharmacological Pathways and Clinical Databases for DHI Prediction

Cheminformatics utilizes computational tools to analyze the chemical properties of compounds and predict their behavior in biological systems [51,52]. Considering that molecules with similar chemical properties may show similar trends as to their action on biological pathways, this approach is particularly useful for predicting DHIs by analyzing the molecular descriptors of drugs and herbal molecules in order to identify potential similarities [53,54,55]. These descriptors include features like chemical structure, molecular size, shape, charge distribution, and lipophilicity, which help predict how these compounds might interact with enzymes, receptors, or transporters [56]. Molecular fingerprints represent the presence or absence of particular substructures within a molecule as binary vectors. These fingerprints can be used to assess the similarity between compounds, which can be translated to similar action or interaction potential [57,58]. For example, the Simplified Molecular Input Line Entry System (SMILES) is a widely used notation in cheminformatics that encodes chemical structures into linear text strings [54,59]. This representation allows computational tools to easily interpret molecular structures for a range of analyses, such as similarity searches, where chemical structures (e.g., drugs or phytochemicals) can be compared to assess their potential interaction based on structural similarity [60,61,62]. Furthermore, as part of chemoinformatic approaches, SMILES can also be used in molecular docking simulations to predict how compounds may bind to specific targets, such as enzymes or receptors, hence identifying potential antagonistic or synergistic effects for the same target [54]. Molecular docking simulations can reveal the synergistic, additive, and multi-target (SAM) effects of herbal medicines by quantifying network interactions, estimating kinetic parameters, and enhancing screening efficiency when combined with pharmacophore modeling [63]. Overall, by encoding both the structural and chemical properties of molecules, SMILES plays a central role in computational approaches in predicting drug interactions and is therefore often incorporated in ML algorithms that rely on molecular descriptors in drug research.

Pharmacological pathways refer to the biological targets, such as receptors, enzymes, or transporters, that a pharmacologically active compound (e.g., a drug molecule) binds and to the physiological responses that this binding produces. Subsequently, pharmacological action refers to the specific biological effects that a drug or herbal compound exerts on the body, often mediated through interaction with a particular target, such as an enzyme, receptor, or transporter [64,65]. In predicting DHIs, pharmacological pathway analysis is important to identify how herbal compounds might interfere with drugs’ actions, either by competing for the same targets or by altering metabolic or signaling pathways [8,66,67].

Clinical information is also essential for accurately assessing the significance of DHIs, as individual factors such as comorbidities, renal and liver function, and pharmacogenomic variations can significantly alter the pharmacological mechanisms involved [68,69,70,71]. For instance, patients with multimorbidity, especially those with severe conditions like cancer or autoimmune diseases, are often treated with drugs that have a narrow therapeutic index (NTI) or require therapeutic drug monitoring (TDM), such as chemotherapy agents and immunosuppressants, where even minor fluctuations in drug levels can lead to toxicity or therapeutic failure [72,73,74]. Elderly patients are another special group, not only because of frequently observed comorbidities but also due to reduced organ function [75,76,77,78]. Reduced renal function, for example, can further alter the PK profile of drugs by reducing excretion, which, in the case of an interaction, may lead to drug accumulation and an increased risk of toxicity [79]. Furthermore, pharmacogenomics also plays a critical role in clinical outcomes. Genetic variations in enzymes like CYP3A4/5, CYP2D6, or transporters can affect how individuals metabolize and transport both drugs and herbal components [80,81]. Patients with genetic polymorphisms may experience faster or slower metabolism, leading to drug levels that differ from those expected. Moreover, genetic differences may also alter drug efficacy, particularly in cases of synergism or antagonism.

In the context of DHIs, integrating data from cheminformatics, pharmacological pathways, and pharmacological action allows a better understanding not only of the potential for interaction but also the mechanism and the clinical outcome [27,82,83,84]. Chemoinformatic allows for the rapid identification of herbal compounds that might interact with drugs based on their chemical structures. Pharmacological pathway analysis provides a deeper understanding of how these interactions occur within the body, mapping the molecular mechanisms that drive DHIs. Finally, understanding the pharmacological actions of both herbal and drug compounds allows for the prediction of the clinical consequences of these interactions, including whether they lead to adverse effects, altered drug efficacy, or even improved therapeutic outcomes under specific conditions.

Several databases compile information on drugs, herbal compounds, and their potential interactions, as presented in Table 1 [85,86,87,88]. These databases provide essential data for building AI models that can be exploited in DHIs studies. They offer a wide range of valuable data, including chemical structures for molecular modeling and predictive analytics, as well as PK and PD information on both drugs and phytochemicals. These data can be utilized in various ways to train AI models for predicting DHIs.code.

## 3. AI Methods for DHI Prediction

AI methods have emerged as a powerful tool for the assessment of DDIs by analyzing different levels of data such as chemoinformatics, pharmacological pathways and clinical parameters [27,82,83,84]. Various AI methods, including traditional ML, DL and other network-based approaches, have been employed to predict DDIs [27,33,34,48,89,90,91]. Since the underlying pharmacology is similar, these approaches can be further extended for DHIs studies. DDI/DHI prediction tasks can be categorized based on the type of learning approach as well as the prediction task. DDI/DHI prediction tasks are typically divided into three categories: binary classification, multi-class classification, and multi-label classification [24,90]. In binary classification, the goal is to predict whether an interaction exists between two drugs or a drug and a phytochemical. In multi-class classification, the task is to predict the specific type of interaction between drug pairs or drugs and phytochemicals, respectively. For multi-label classification, the objective is to predict multiple interaction types when two or more interactions occur between the same drug pair. Supervised learning is often applied to binary, multi-class, and multi-label classification tasks, where labeled data are available to train models to predict whether interactions exist, the specific types of interactions, or multiple interaction types between drug pairs. Unsupervised learning, on the other hand, does not rely on labeled data and is typically used for discovering hidden patterns or clusters of drugs and herbs based on their similarities. Semi-supervised learning combines both labeled and unlabeled data, making it useful in scenarios where labeled interaction data are limited but unlabeled data can help improve prediction performance [24,90].

### 3.1. Traditional Machine Learning Approaches

ML models are often classified based on the way they handle labeled data, which defines the nature of the learning process. ML methods, including supervised, unsupervised, and semi-supervised learning, are widely used for predicting DDIs. These methods can potentially be applied to DHIs as well. In the following, we focus on traditional ML approaches within each type of learning that could be applied in DHI prediction. Subsequently, we will explore deep learning (DL) methods, categorized by their data representation approach, indicating a complementary perspective. We will illustrate the types of learning methods with examples showing how they could be applied to DHIs based on available data patterns.

#### 3.1.1. Supervised Learning

Supervised learning (classification) involves training a model on a labeled dataset where the outcome (e.g., the presence or absence of an interaction) is known [92]. For DHI prediction, supervised learning models can be trained on datasets containing known interactions. Features such as molecular fingerprints, structural descriptors, and pathway information can be used as inputs. Random Forest (RF) is an ensemble learning method that builds multiple decision trees and merges them to obtain a more accurate and stable prediction. In predicting DHIs, RF models can manage large numbers of features derived from chemoinformatic data and identify which features are most important for predicting interactions. Support Vector Machines (SVMs) are particularly effective for high-dimensional data and can be used to classify drug–herb pairs as interacting or non-interacting based on their chemical properties and other features. Gradient Boosting Machines (GBM models, such as XGBoost), build an ensemble of decision trees sequentially, each one correcting the errors of its predecessor. GBM models are highly flexible and can be fine-tuned to predict DHIs with high accuracy.

A predictive model for DHIs can use supervised learning methods, such as Random Forest (RF), to explore interactions like those between St. John’s Wort (SJW) and cyclosporin. Hyperforin induce the metabolic enzyme CYP3A4 and P-gp transporter leading to increased metabolism and potential subtherapeutic levels of drugs like cyclosporin. This interaction is clinically significant, as it can reduce cyclosporin’s effectiveness and potentially lead to therapeutic failure. An RF model trained on a dataset of known DHIs could use features such as the molecular descriptors of herbal compounds, enzyme induction potential, and the chemical structure of drugs to predict the likelihood of an herb impacting CYP3A4 and P-gp substrates like cyclosporin. Given the limited availability of DHI-specific data, such models can also leverage DDIs datasets, where enzyme modulation mechanisms are well-documented. By incorporating pathway and mechanistic data, an RF model can expand its predictions from DDIs to DHIs, helping identify potential risks of therapeutic failure. This approach underscores the importance of integrating mechanistic insights, pathway information, and chemical descriptors for accurate DHI predictions and proactive clinical management [93,94,95].

#### 3.1.2. Unsupervised Learning/Dimensionality Reduction

Unsupervised learning methods are useful when labeled data are unavailable. Clustering Algorithms such as k-means or hierarchical clustering, can group drugs and herbs based on their chemical similarity or other relevant characteristics, such as their targets [96,97]. The resulting clusters can highlight herb compounds that are structurally similar to known drug perpetrators, suggesting potential DHIs. Dimensionality reduction techniques, such as Principal Component Analysis (PCA) or t-SNE (t-Distributed Stochastic Neighbor Embedding) can further simplify complex chemoinformatic data by reducing its dimensionality, eliminating redundant data properties while preserving key information [98,99]. This helps visualize patterns and relationships that may indicate interactions. Together, these methods help uncover potential interactions by identifying patterns of similarity that may indicate common pathways or mechanisms of action, which is especially valuable in cases where direct interaction data are limited.

An example of an unsupervised approach could be based on the well-studied grapefruit juice which is known to inhibit CYP3A4, especially in the intestinal wall, leading to increased bioavailability of drugs like simvastatin that lead in increased drug’s concentrations in the blood stream and eventually raising the risk for adverse events and toxicity such as rhabdomyolysis in the case of simvastatin [100,101,102,103]. Using unsupervised clustering techniques, herbs can be grouped based on their ability to inhibit CYP enzymes or affect drug transporters (like P-glycoprotein) involved in drug absorption [104,105]. Grapefruit would cluster with other herbs that share similar effects, predicting a potential interaction with drugs metabolized by these pathways. This method could help in identifying lesser-known herbs that might similarly affect the absorption of per-os administered drugs [106,107,108,109,110]. A similar approach, based on relation clustering, has already been proposed to address the lack of data in the problem of food–drug interaction (FDI) extraction [96].

#### 3.1.3. Semi-Supervised Learning

Semi-supervised learning combines elements of supervised and unsupervised learning. It is particularly useful when there is a small amount of labeled data (known interactions) and a large amount of unlabeled data (unknown interactions). In self-training, a model is first trained on the labeled data. It then makes predictions on the unlabeled data, and the most confident predictions are added to the labeled dataset for further training. This iterative process can improve the model’s ability to predict DHIs by leveraging both labeled and unlabeled data. For example, ginkgolide B from ginkgo biloba has been reported to affect the distribution of drugs by modulating drug transporters like P-gp, potentially altering the blood–brain barrier (BBB) permeability of drugs in CNN [111]. A semi-supervised model could be trained using a small dataset of known interactions affecting drug distribution or BBB transport. The model can then predict potential interactions between phytochemicals similar or alike ginkgolide B and CNS drugs that are substrates to specific transporters (i.e., benzodiazepines) by identifying similar patterns in herb-induced modulation of transporters that affect drug distribution. The model can iteratively improve its predictions by incorporating new data from both labeled and unlabeled sources.

### 3.2. Deep Learning Methods

DL is a branch of ML that leverages multiple layers to extract features from raw data, enabling the identification of complex patterns relevant to the input. Its techniques include Convolutional Neural Networks (CNNs), Recurrent Neural Networks (RNNs), and Deep Neural Networks (DNNs), and it can be applied across unsupervised, semi-supervised, and supervised learning frameworks. DL methods have proven highly effective in drug research due to their ability to learn complex, non-linear relationships in large, high-dimensional datasets [37,89,112]. By incorporating data from multiple sources, such as chemical structures, pharmacological profiles, and pathway information, deep learning models can uncover hidden patterns and provide more accurate predictions, making them invaluable tools in modern drug discovery and safety evaluation [113]. These models vary based on how the input data are represented, leading to different approaches tailored to specific types of drug-related data. For instance, CNNs are often used to process drug molecular structures represented as images, while graph-based neural networks (GNNs) excel at representing drugs and their interactions or atoms and their chemical bonds in graph form. Knowledge-graph-based models integrate domain-specific knowledge about drugs, pathways, and targets. Hybrid and multi-modal approaches combine various data sources to improve predictive accuracy (Figure 2) [114]. In this context, the choice of deep learning method depends on the nature of the data being used, whether it involves molecular structures, biological pathways, or sequential drug usage patterns. Below, we explore the most common deep learning techniques for DDI/DHI prediction, categorized based on how they handle and process input data.

#### 3.2.1. Deep Neural Networks (DNNs)

Traditional Deep Neural Networks (DNNs) are composed of numerous interconnected neurons, typically organized into an input layer, multiple hidden layers, and an output layer. In DDI prediction, drugs often undergo transformations or feature extraction before being input into the network (Figure 2) [115]. For DHI prediction, DNNs can be trained on chemoinformatic data, such as molecular fingerprints, to predict interactions. For example, curcumin, the active compound in turmeric, has anti-inflammatory properties and may interact with the primary target of aspirin, cyclooxygenase-2 (COX-2) enzymes, potentially potentiating its efficacy [116]. A DNN could be trained on a dataset containing drug-target interactions, including enzymes and receptors or other targets. By incorporating molecular fingerprints of both the herb and drug, the DNN can predict potential interactions at the COX-2 enzyme, suggesting whether curcumin or similar phytochemicals could enhance or inhibit the anti-inflammatory effects of aspirin or other drugs that target COX-2 enzymes.

CNNs use convolutional filters to extract relevant local features from data. After passing through several convolutional layers, these networks calculate the probability of a DDI. CNNs have also been applied on drug–drug similarity matrices, though their lack of spatial information means CNNs must be applied cautiously [34,39,117]. Additionally, while CNNs are well suited for grid-like data, such as images, representing molecular structures as images may be less effective than other approaches, like graph-based representations, which more accurately capture molecular connectivity and relationships. In chemoinformatics, CNNs can be used to analyze molecular structures represented as 2D or 3D grids, identifying patterns that indicate potential interactions between drugs and herbs. Green tea contains catechins which can interact with several cardiovascular drugs, such as nadolol by either affecting drug transporters or secondary receptors involved in its pharmacodynamics [118]. CNNs could be used to analyze the 3D molecular structures of catechins in green tea and their similarity to known ligands for nadolol’s PK or PD targets. By processing these structures, the CNN can predict whether green tea or other herbal products that contain catechins might affect nadolol’s action pharmacological efficiency and potentially lead to altered therapeutic outcomes.

Recurrent Neural Networks (RNNs) are designed to manage sequential data, making them suitable for text mining as well as for modeling interactions, such as the effect of drug–herb co-administration over time [119,120]. RNNs also excel in analyzing time-dependent data, which can be valuable in PK/PD modeling [121]. However, these applications remain highly underexplored. Long Short-Term Memory (LSTM) networks, a type of RNN, can learn long-range dependencies, making them useful for predicting delayed or cumulative interactions. In this case, LSTMs can be applied to study potential DHIs with mechanisms similar to those of SJW which in acute co-administration may raise the concentrations of some drugs due to inhibition in CYPs, but long-term administration may lead in sub-therapeutic concentrations due to induction of abundance of CYP enzymes in the liver [21].

#### 3.2.2. Graph-Based Approaches

Graph-based approaches model the relationships between entities (e.g., drugs, herbs, targets) as graphs, where entities (like atoms or molecules/drugs) and edges represent interactions (like chemical bonds or interactions among drugs) [122,123,124]. These methods are particularly useful for integrating diverse types of data and predicting novel interactions [125,126,127,128]. Expanding these models to incorporate additional entities, such as targets, pathways, or diseases, advances the framework into knowledge graph (KG) representation. In KGs, entities are expanded to include drugs, genes, pathways, and diseases, while relationships indicate various interactions among these entities, such as binding, inhibition, activation, or transport.

Link prediction algorithms, such as random walks or matrix factorization, can be used to predict new interactions by analyzing the structure of the network [129]. These algorithms identify pairs of nodes that are likely to be connected based on their proximity or similarity within the network. For example, garlic can affect the metabolism and bioavailability of saquinavir, reducing its plasma concentrations and antiviral efficacy [130]. A drug–herb interaction network could be constructed where nodes represent drugs, herbs, and metabolic enzymes (i.e., CYP3A4). By analyzing the network structure and using link prediction algorithms, it could predict potential interactions between garlic and saquinavir, particularly focusing on how the induction of metabolic pathways by garlic may reduce saquinavir’s effectiveness.

Another approach is through Graph Neural Networks (GNNs), which are a class of neural networks designed to operate on graph-structured data [131,132,133,134]. GNNs can learn representations of nodes and edges that capture the complex relationships between drugs and herbs, enabling the prediction of novel interactions [134]. An example in this case is baicalin—an active ingredient of Chinese herbal medicine Scutellaria baicalensis— which is known to inhibit organic anion transporters (OATPs), which play a role in the kinetics of several drugs, e.g., statins, with evidence suggesting a potential impact on the PK profile of rosuvastatin [135]. A GNN could be employed to model the interactions between herbs, drugs, and transporters involved in drug elimination. By representing these interactions as a graph and training the GNN, the model can predict whether herbs with active ingredients such as baicalin might inhibit OATPs, leading to PK profiles of drug substrates such as statins and potentially increasing the risk of adverse effects or reduced therapeutic action.

#### 3.2.3. Natural Language Processing and Text Mining

##### Text Mining

Natural language processing (NLP) techniques can be applied to extract information on DDIs, as well as on DHIs, from unstructured text sources, such as the scientific literature, electronic health records (EHRs), and drug labels (Figure 3) [136,137]. Especially for DHIs, recently, an approach utilizing NLP algorithms was introduced to assist researchers and readers in quickly identifying key information from clinical studies and case reports on DHIs, enhancing the efficiency of literature interpretation [138]. Text mining involves extracting structured information from unstructured text. The text is transformed into vectors through several techniques such as one-hot encoding or text frequency inverse document frequency (TF-IDF) and further processed through ML approaches [139]. In the context of DHIs, text mining can identify mentions of interactions, herb constituents, and pharmacological effects from large corpora of text. One of the popular approaches is through Named Entity Recognition (NER) models, which can identify entities such as drugs, herbs, and interactions within text. By analyzing co-occurrences of these entities, it is possible to infer potential DHIs. A relative approach is that of relation extraction models that identify and classify the relationships between entities. For example, these models can extract statements indicating an interaction between a specific drug and herb based on the existing scientific literature. For example, kava is considered to have psychoactive effects and may enhance the sedative effects of benzodiazepines, potentially leading to excessive sedation or respiratory depression [140]. An NLP such as NER or relation extraction could be used to mine the scientific literature and clinical case reports for mentions of interactions between kava or relative herbs with psychoactive effects on the CNS and benzodiazepines or other CNS drugs. By identifying and classifying these interactions, NLP can extract valuable insights into how these herbs might modulate the pharmacodynamics of CNS drugs such as sedatives, potentially enhancing their pharmacologic effects, even if the phytochemical composition is different among the herbs.

##### Knowledge Graph Construction

Knowledge graphs (KGs) represent relationships between entities as a graph, where nodes represent entities and edges represent relationships. NLP can be used to construct knowledge graphs from text by extracting entities and their relationships. KGs like those built from resources such as MEDLINE or EHRs can be used to link drugs, herbs, targets, and interactions. These graphs can then be mined to predict novel DHIs by identifying paths between drugs and herbs. For example, KGs can be applied to study DHIs, such as ginseng’s impact on the metabolism of warfarin through its impact on CYP enzymes, potentially reducing warfarin’s anticoagulant effect [141]. NLP can be used to build a KG by extracting data on pharmacokinetics (e.g., absorption, distribution, metabolism, and elimination) of ginseng and warfarin from various text sources. The knowledge graph can represent these interactions, linking herbs, drugs, enzymes, and transporters. By querying the graph, potential interactions where ginseng might reduce warfarin’s anticoagulant effect can be identified.

#### 3.2.4. Generative Modeling-Based Approaches

Generative modeling involves learning the data’s underlying distribution with the intent of generating new samples. They generate outputs such as images, text, or audio that resemble real-world examples. By modeling complex data distributions, generative models can produce realistic and diverse content, often used for tasks like image synthesis, text generation, and even drug discovery. The application of such models in molecular discovery was analyzed in a recent review showing a wide applicability in conditional molecule generation and property description such as toxicity from SMILES [142,143].

##### Variational Autoencoders

Variational Autoencoders (VAEs) are used to learn latent representations of molecules, allowing for the generation of new molecules by sampling from this learned distribution. They create a smooth latent space, which enables interpolation even with discrete entities like molecules. VAEs can be adapted for modeling chemical molecules by using compatible encoders and decoders, often leveraging string inputs such as SMILES. The latest models include adding condition vectors to the input, enabling generation tailored to specific conditions [144]. VAEs have since expanded to multi-modal applications, incorporating contextual data like gene expression or protein targets for task-driven molecule generation. Reinforcement learning can be applied to guide VAEs in producing molecules with specific properties by using reward signals.

A potential example of using VAEs in DHIs could involve DHIs of herbs with antiplatelet or anticoagulant therapies. For example, ginger may potentiate the effects of anticoagulants, such as warfarin, possible via the inhibition of thromboxane synthetase, although data are scarce or conflicted, maybe due to the differences in phytochemical content or among supplements [145,146]. A VAE model could be trained on a dataset of DDIs/DHIs, encoding molecular features and PK/PD data. For instance, input features could include molecular descriptors, pathway data, and structural information about active compounds in ginger and warfarin or other coumadin analogs. The VAE would learn latent representations capturing the interactions’ variability, allowing it to generate new, realistic data points on the potential interaction space between ginger compounds and warfarin.

##### Transformers

A transformer is a neural network model initially designed for machine translation tasks. It consists of an encoder that processes a sentence in the source language and a decoder that sequentially generates the translation in the target language. The model’s strength lies in its attention layers, which capture the contextual meaning of words and sub-words. By splitting text into sub-words, the transformer encodes these sequences and uses attention to link relevant information, allowing it to generate an accurate translation through its decoding component. A prime example of transformers’ applications is their pivotal role in AlphaFold2, a leading system for predicting the 3D structure of proteins [147]. For generative modeling, decoder-only transformers like the Generative Pre-Training Transformer (GPT) [148] have become the dominant approach. MolGPT was one of the first models to leverage the transformer architecture for conditional molecule generation [149]. It processes SMILES tokens combined with a condition vector representing desired molecular properties and scaffolds, training on a next-token prediction task to generate new molecules. GPT-based models, paired with reinforcement learning, can also optimize molecular properties, such as pIC50, by learning embeddings from SMILES strings and adjusting the embedding space to generate molecules with specific traits. Expanding on GPT-like approaches, the regression transformer reimagines conditional sequence modeling as a regression task, creating a multitask model that predicts properties and generates molecules by combining molecular and property tokens, using an alternating masking scheme during training. The ability of transformer-based language models to parse and interpret complex scientific literature sources and extract information about known DDIs along with pharmacological and or clinical data is of great help to automatically aggregate relevant data from diverse sources [139,150,151,152,153]. This can further enable the assessment of potential DHIs related to phytochemicals that are unknown or inexplicitly reported. This type of generalization refers to a model’s ability to accurately predict interactions between drugs and herbs that may not have been directly encountered during training. This is possible by capturing underlying patterns in molecular structures, biological pathways, and PK properties. This ability comes from embedding drugs in a high-dimensional space, where proximity often correlates with pharmacological similarity, enabling models to suggest interactions based on structural or functional analogs. Moreover, the attention mechanisms in transformer-based language models allow them to generalize by understanding the contextual roles of drugs in various pathways and scenarios. This can help the model infer interactions based on the broader biological and chemical context, which is essential when considering drugs with different mechanisms of action that may intersect in unexpected ways. On the other side, such models may struggle with out-of-distribution generalization—predicting interactions for novel drugs with unique structures or mechanisms not well represented in the training data. If certain drug types or interactions are underrepresented, the model may generalize poorly, especially in rare or novel interaction types. This is a significant consideration when introducing new types of molecules, i.e., phytochemical from herbal remedies with unknown interaction profiles.

A transformer-based language model, such as BERT, can be used to predict DHIs by analyzing extensive data from experimental studies, clinical trials, and scientific literature. For example, considering the interaction between rhodiola rosea (RR), an adaptogenic herb known for its stress-reducing and endurance-enhancing benefits, and losartan, an angiotensin II receptor blocker used to treat hypertension, the model could process data on how RR might influence the pharmacokinetic (PK) profile of losartan through modulation of CYP2C9 mediated metabolism [154,155]. The model could analyze PK and pharmacodynamic (PD) data to identify patterns suggesting that RR could alter losartan’s PK, potentially impacting its antihypertensive efficacy or the action of other drugs that are substrates to CYP2C9. Additionally, RR’s adaptogenic effects and its potential influence on blood pressure regulation could further modulate losartan’s or other antihypertensive drugs’ PD effects or the effects of drugs that act on CNS [156,157]. By training on datasets that include both pharmacological pathways and clinical outcomes, a transformer model could predict the likelihood of a DHI, such as a PK-related interaction or a PD-related hypotensive effect, helping clinicians assess potential risks hence guide towards the safe use of RR in combination with cardiovascular medications.

### 3.3. Hybrid and Integrative Approaches

Integrative approaches combine multiple AI methods and data sources to improve the prediction of DHIs. These approaches can leverage the strengths of different techniques to provide more accurate and comprehensive predictions.

#### 3.3.1. Multiview Learning

Multiview learning involves combining different types of data (views) to improve predictive performance [158,159]. This can be achieved through co-training where two or more models are trained on different views of the data. Each model’s predictions are used to iteratively label the unlabeled data for the other models, improving the overall performance. Alternative ensemble models can be applied that combine the predictions of multiple models to produce a final prediction. For DHI prediction, ensemble models can integrate predictions from different types of models (e.g., SVMs, RFs, GNNs) trained on different data sources, such as chemoinformatic data along with pharmacological pathway information. For example, furanocoumarins in grapefruit juice can inhibit drug-metabolizing enzymes and transporters, affecting the absorption and metabolism of drugs like statins or other substrates [100,101,102,103,104,105]. Multiview learning could be used to integrate different data views, such as the inhibition potential of furanocoumarins on drug-metabolizing enzymes and transporters, along with the chemical properties of drug substrates. By combining these views, a model can predict how grapefruit juice or other similar herbals, or food supplements might alter drugs’ PKs; i.e., atorvastatin’s or similar molecules could result in increased plasma levels and in a higher risk of toxicity [106,107,108,109,110].

#### 3.3.2. Transfer Learning

Transfer learning involves applying knowledge “learned” from one domain to another [92,160]. In the context of DHI prediction, models trained on well-studied DDIs can be adapted to predict DHIs, especially when data for certain herbs are sparse (Figure 4). Transfer learning can employ domain adaptation techniques that adjust a model trained in one domain (e.g., drugs) to work in a different but related domain (e.g., herbs). An alternative approach is that of pre-trained models, such as those developed for NLP tasks which can be fine-tuned on domain-specific data to extract relevant information from text related to DHIs. An application example of drug–herb interaction using transfer learning could involve training a model on a well-studied dataset of DDIs to leverage its learned knowledge for predicting DHIs. For instance, a model trained on interactions involving statins and various drugs could transfer its learned features to predict interactions between specific statins (i.e., simvastatin, lovastatin or atorvastatin) and herbal products like red yeast rice, which contains natural statins (i.e., monacolin K). This approach allows the model to recognize interaction patterns involving synergism in this example due to similar chemical structures and action since all statins function as HMG Co-A reductase inhibitors (3-hydroxy-3-methyl-glutaryl-coenzyme A reductase). That knowledge can be applied to novel scenarios where labeled data for DHIs might be scarce, e.g., herbs with constituents like statins that in turn could also function as HMG Co-A reductase inhibitors [161]. Another example could be for phytochemicals that may interfere with the therapeutic drug monitoring process as another type of DHI. For example, ashwagandha, which is often applied in stress situations, may affect the PK/PD profile of digoxin due to interferences in the bioanalytical method that quantifies the drug, potentially altering digoxin’s quantitation [162]. Transfer learning can be employed to adapt models trained on extensive data for well-studied DDIs to predict interactions between digoxin and less studied herbs like ashwagandha. For example, a model trained on drugs involving digoxin and related interferences on quantitation mechanisms can be fine-tuned to predict whether herbs such as ashwagandha can modulate the TDM measurements, helping to identify potential risks or synergies.

## 4. Explainable AI (XAI) in DHIs

Explainable AI (XAI) is gaining attention in drug research, as it addresses the need for transparency when complex AI models are exploited [163]. XAI refers to AI models and algorithms designed to provide clear insights into how AI predictions and decisions are made. Unlike traditional AI models, which often function as “black boxes” by producing results without revealing their reasoning, XAI prioritizes transparency, interpretability, and accountability. Consequently, XAI enhances trust and reliability in critical tasks like DDI prediction by revealing the underlying decision-making processes [45,164,165].

Mechanistically, XAI models achieve interpretability through various techniques. Methods such as SHAP (Shapley Additive Explanations) and LIME (Local Interpretable Model-agnostic Explanations) highlight which input features—such as chemical structure, PK or PD properties—exert the most influence on predictions [166,167]. These methods break down complex model predictions by attributing each feature’s contribution to the outcome, helping identify key input features. Attention mechanisms can also enhance explainability by identifying salient features in DDI predictions [168]. Additionally, meta-path-based information fusion and attention mechanisms are used to capture complex relationships among drug-related biological entities, partially explaining predicted DDIs [169]. Visualization techniques like attention maps can offer accessible insights into interaction pathways or relevant biochemical factors [45]. Tree-based models are increasingly supported by tools that provide a global understanding by combining local explanations, enabling the identification of non-linear risk factors and interaction effects [170]. In addition to standard XAI techniques, biologically inspired neural networks have been proposed to enhance model interpretability by mimicking hierarchical biological structures and processes [171,172]. These models simulate complex biological systems more closely, offering insight into mechanisms underlying drug response, drug synergy, and disease progression. By structuring neural networks to reflect pathways and cellular interactions, researchers can align AI model outputs with biological knowledge, fostering greater confidence and transparency in predictions. Leveraging such biologically inspired models for DDI and DHI prediction could improve interpretability by connecting predicted outcomes to realistic biological interactions and mechanisms.

Currently, the applicability of XAI techniques is being explored for pharmacovigilance studies to investigate drug treatments, side effects, and interactions using various data sources and model types [173]. Hence, XAI could be particularly useful for understanding the complex nature of herbal products, with XAI models able to leverage structured chemoinformatic data on drug categories as well as phytochemical compounds, metabolic enzymes, transport proteins, and therapeutic targets to categorize interactions and highlight risks for specific drug types—especially NTI or TDM drugs, such as antidepressants, immunosuppressants, anticancer agents, and anticoagulants. For example, SJW presents multifaceted risks for DHIs through both PK and PD mechanisms, particularly with drugs that are substrates of P-glycoprotein (P-gp) transport and CYP3A4/5 metabolism, or drugs that act on the central nervous system. By applying SHAP values within an XAI model for DHI, it becomes clear that hyperforin’s induction of CYP3A4/4 and P-gp has a significant impact on reducing the concentrations of CYP3A4/5 and P-gp substrate drugs, such as cyclosporine. This interpretability allows clinicians to understand why there is a high risk of subtherapeutic concentrations and highlights the molecular pathways involved, helping guide healthcare providers to monitor or adjust therapeutic plans, or even advise against using SJW. For PD mechanisms, the components hypericin and hyperforin in SJW are known to increase serotonin levels in the brain by modulating monoamine oxidase (MAO) activity, potentially leading to serotonin syndrome when combined with drugs that act similarly, such as selective serotonin reuptake inhibitors (SSRIs). However, the risk is not limited to SSRIs; other drug classes that influence serotonin, such as tricyclic antidepressants (e.g., amitriptyline), monoamine oxidase inhibitors (e.g., phenelzine), and certain migraine medications (e.g., triptans), could also interact with SJW. Using XAI, an AI model can analyze patient medication regimens across these drug classes, assigning attention scores or feature attributions to pinpoint precisely how SJW affects serotonin activity in combination with each drug type, even for drugs with indirect impacts on serotonin levels. Clinicians can then recognize the multifaceted risk of serotonin syndrome with SJW, involving multiple serotonin-modulating drug classes, and take appropriate actions to safeguard patient safety.

## 5. Mapping AI Models to Data Types for Assessment of DHIs

Regardless of whether a DDI or DHI approach is aimed at, the AI models presented can be leveraged according to their specific strengths and suitability for different analytical tasks. Each model’s strengths should align with the demands of various data types, creating a versatile framework for comprehensive interaction analysis. Depending on data availability, specific methods can be matched to different data types (chemoinformatic data, pharmacological pathways, or clinical data) based on their richness and accessibility (Figure 5). The careful selection of AI models according to data type and availability enables a comprehensive and adoptive approach to drug interaction analysis, maximizing insights and improving prediction accuracy across different contexts, such as the case of DHI. This could further guide either validation through targeted experimental procedures or adaptation within clinical decision support systems towards ADR risk minimization. Experimental methods that could complement AI predictions range from in vitro, microsomal, and cell culture assays to in vivo animal experiments, physiologically based models, and even clinical trials. These approaches could consider AI-driven results for single-ingredient herbs, crude extracts, or standardized herbal formulas, collectively building evidence-based data on the potential clinical significance of DHIs for clinical utilization [154,155,157,174,175,176].

Chemoinformatics approaches primarily benefit from models like supervised learning, DNNs, GNNs, and generative models such as VAEs and transformers. These models, including hybrid approaches like transfer learning, are well equipped to interpret chemical structures and predict DDIs/DHIs based on molecular properties. If pharmacological pathway data are available for DHI analysis, then supervised, unsupervised, and semi-supervised learning methods, as well as GNNs, DNNs, generative models, and hybrid approaches, can be applied effectively. These models excel at analyzing complex biological pathways and can provide insights into systemic drug effects. In cases where clinical data—such as patient outcomes, genotype/phenotype information, and disease characteristics—are available, models like supervised learning, NLP, DNNs, and hybrid approaches fit better. These tools have the potential to excel at analyzing real-world data from clinical settings to assess potential DDIs and, by extension, DHIs.

Considering the complex nature of herbal products, unsupervised and semi-supervised learning have broad applications across all data types, particularly for DHI analysis. Unsupervised learning is useful when no labeled data are available, such as when exploring new, uncharted interactions or identifying clusters of similar drugs, side effects, or interactions in available data. This approach is especially beneficial for combinational datasets, such as those involving both chemoinformatics and pharmacological data. Similarly, semi-supervised learning is effective when a small, labeled dataset exists (e.g., known drug interactions) alongside a larger unlabeled dataset. This approach enhances generalization by leveraging both labeled and unlabeled data, enriching datasets where pharmacological pathway or clinical data may be incomplete.

## 6. DHI and AI Tools Under the Prism of Network Pharmacology, Systems Biology, and Computational Toxicology

In the context of state-of-the-art approaches in drug research, the integration of AI approaches within network pharmacology, systems biology, and computational toxicology offers transformative tools for predicting and understanding DHIs. Regardless of the approach, the exploitation of AI tools can provide a robust framework for integrating diverse data types and generating insights into drug interactions, systemic effects, and toxicity profiles that can be further exploited.

Network pharmacology provides a detailed map of multi-compound and multi-target interactions and can incorporate several levels of data considering chemoinformatics or other pharmacological relative information (i.e., inhibitor, inducer, competitive, non-competitive binding, etc.) [177]. Within a network pharmacology analysis, these pharmacological pathways describe the series of biological outcomes that occur within a biological system in response to a drug or herbal compound [178]. By mapping the targets of a drug and an herb onto a biological network, it is possible to predict potential interactions based on shared or interconnected pathways. Hence, in the context of DHIs, network pharmacology would benefit from AI models like GNNs for representing drug–target, herb–target, drug–pathway and herb–pathway interactions; supervised learning for predicting interactions; generative modelling; and hybrid models for integrating various data sources. Especially for herbs, this can be the summary of all compounds that are known to have any kind of pharmacological action. This network-based approach allows for the identification of not only direct DHIs but also secondary interactions that might arise from downstream effects in the metabolic or signaling pathways. A typical example can be mentioned again: that of SJW, with its multipotent actions on PK and PD processes and the drugs that are sharing them.

Systems biology focuses on understanding the complexity of biological systems by integrating data from multiple levels, such as genomics, proteomics, metabolomics, and transcriptomics [179,180]. When applied to DHIs, systems biology allows for a more comprehensive view of how herbal products and drugs interact within the context of the entire biological system [181]. By applying systems biology principles, AI models can predict how an herbal product might influence not only the primary metabolic enzymes (like CYPs) but also secondary processes such pharmacodynamic-related receptor signaling. Hence, AI-driven systems biology models can simulate how herbal compounds influence PK or PD processes across various organs and tissues by leveraging, for example, DNNs and GNNs for modeling complex biological networks and drug effects at the system level, and unsupervised learning to uncover hidden relationships and patterns in multi-omics data. This system-level approach can identify potential DHIs that traditional single-target models might omit. Furthermore, it provides deeper insights into how DHIs can appear in clinical settings, improving the ability to predict adverse outcomes or therapeutic failures. Moreover, it can incorporate patient-oriented genotype/phenotype data that allow the generation of more personalized predictions.

In addition to network pharmacology and systems biology, it is also critical to identify DHIs that should be strictly avoided due to the risk of toxic effects [182]. Such cases often involve drugs with a narrow therapeutic index, which typically require therapeutic monitoring as any unexpected change in their PK or PD profile can have a significant impact on clinical outcomes. Drugs like warfarin (with a target international normalized ratio of 2.0–3.0 for most indications), digoxin (0.5–2.0 ng/mL), theophylline (10–20 μg/mL), and cyclosporine (150–400 ng/mL) are typical examples where special precautions should be taken to prevent alterations in their therapeutic effects and avoid toxicity. In this context, computational toxicology offers crucial insights into the risks of adverse drug reactions (ADRs) or toxic effects [183]. Most computational toxicology models leverage chemoinformatics approaches to predict the toxic effects of chemical compounds. This is particularly relevant for predicting adverse DHIs, where the combined effects of a drug and an herb may lead to toxicity. ML models can be employed to predict the toxicity of drug–herb combinations by analyzing chemoinformatics features, structure–activity relationships (SARs) associated with toxic pharmacophores, and pathway information for both drugs and phytochemicals. In this context, we can utilize supervised learning for toxicity prediction, GNNs for pathway analysis, NLP for text-mining adverse effects from clinical records, and generative models to explore safer drug design. These models can identify combinations likely to increase the risk of adverse effects based on their impact on key toxicological pathways.

## 7. Ethical and Practical Challenges of AI in DHI Prediction

AI-driven DHIs predictions face multiple challenges, spanning practical, ethical, and technical domains. One major limitation is the scarcity of high-quality labeled datasets, which affects model performance and generalizability [84,184]. Semi-supervised learning, transfer learning from DDI datasets, and data augmentation techniques can help improve model robustness despite limited data availability. The complexity of multi-component herbal interactions presents another challenge, as herbs contain multiple bioactive compounds that may interact synergistically or antagonistically, making it challenging for standard AI models to capture these dynamics. Multi-modal AI models that integrate molecular, biological, and clinical data can provide a more comprehensive understanding of DHIs. Furthermore, the lack of clinical validation and physician adoption remains a significant barrier, as AI-driven predictions often lack sufficient experimental and real-world validation. Prospective clinical studies, in vitro assays, and multi-omics approaches can help validate computationally predicted DHIs for clinical utilization. On the other hand, as previously stated, AI can guide the design of experimental approaches that focus on key mechanisms and pathways. Finally, successful integration into healthcare systems is crucial for AI-based DHI tools to be practically useful; however, this goal remains challenging, at least in terms of enhancing usability and effectiveness in healthcare settings. A key step forward would be the incorporation of AI systems into clinical decision support systems in an interoperable manner. Even simple AI-generated prompts could help treating physicians exercise caution when considering patient treatment options [185].

Beyond practical concerns, AI-based DHI prediction also raises ethical challenges. Data bias and health inequities arise when AI models rely on training data that lack demographic, pharmacogenomic, and multimorbidity diversity, leading to inaccurate predictions for underrepresented populations [186]. To mitigate this, diverse datasets, fairness-aware algorithms, and bias-detection frameworks should be employed. The lack of herbal regulation and standardization further complicates AI predictions, as variability in sourcing, preparation, and administration can lead to misleading outputs. Standardized herbal databases, developed in collaboration with regulatory bodies, are essential for improving prediction reliability [187]. Additionally, transparency and explainability in AI models remain a concern, as many deep learning approaches operate as “black boxes,” making it difficult for clinicians to trust AI-generated predictions. Implementing explainable AI (XAI) techniques, such as feature importance scoring and knowledge graph visualizations, can enhance transparency and adoption in clinical practice [45,167,173,188]. Patient privacy and data security are also critical, as AI models integrating pharmacogenomic, multimorbidity, and demographic data must comply with regulations such as HIPAA and GDPR. Secure data-sharing techniques, including federated learning and differential privacy, should be utilized to protect sensitive patient information.

## 8. Conclusions

AI technologies hold great promise in the identification and understanding of DHIs, leveraging diverse data sources and molecular analysis to uncover the intricate mechanisms involved. This knowledge provides valuable insights into potential risks associated with drug–herb interactions, facilitating optimal therapy management. The integration of XAI is especially valuable, as it allows researchers and healthcare providers to understand and interpret AI-driven predictions, fostering transparency and trust in AI-assisted recommendations. For example, AI-powered decision support systems in the future could offer personalized recommendations and alerts tailored to individual patients, considering their specific medication regimens and herbal product use. However, AI should complement rather than replace clinical expertise. The effective application of AI in predicting DHIs relies on combining these tools with personalized approaches, evidence-based guidelines, and open communication between healthcare providers and patients about herbal product use. This combined approach will enable safer, more informed utilization of both conventional medications and herbal products, ultimately enhancing patient care.

## Figures and Tables

**Figure 1 pharmaceuticals-18-00282-f001:**
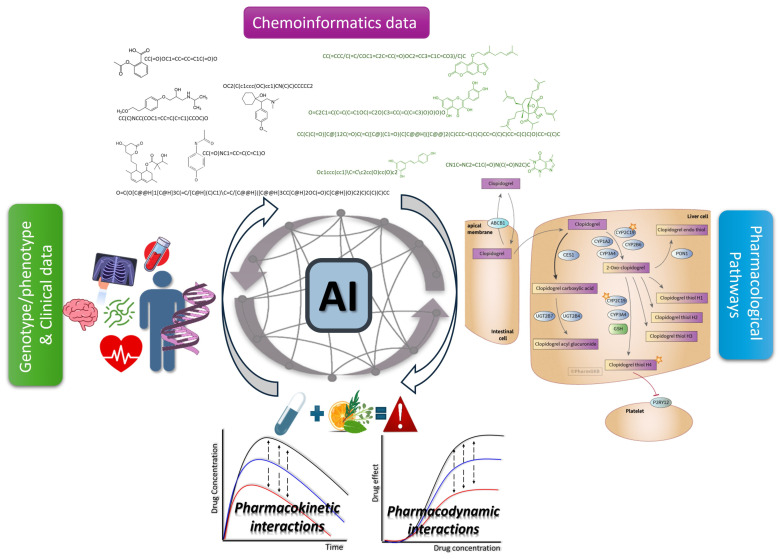
Integration of different types of data through AI tools to predict and evaluate the clinical significance of potential DHIs. (The pharmacological pathway of clopidogrel was adopted from PharmKGB [42]).

**Figure 2 pharmaceuticals-18-00282-f002:**
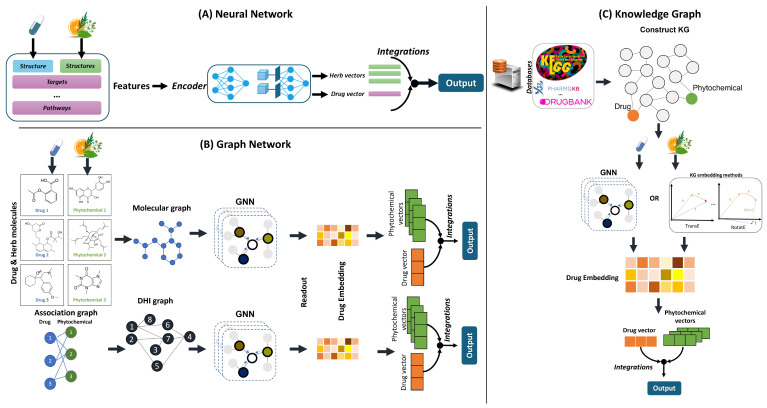
Deep learning methods for assessing DHIs based on the approaches on DDIs (based on ref [99] with modifications).

**Figure 3 pharmaceuticals-18-00282-f003:**
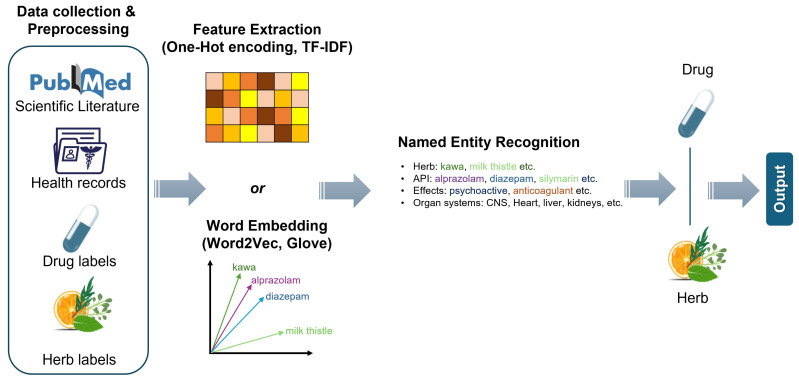
NLP general pipeline of extracting information from unstructured resources to create structured relationships for DHIs.

**Figure 4 pharmaceuticals-18-00282-f004:**
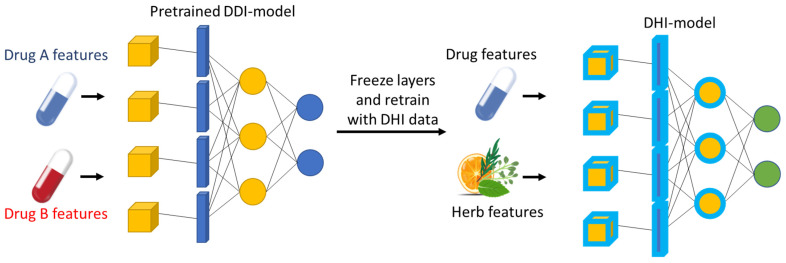
Transfer learning pipeline representing how “knowledge” learned from one domain is exploited in a similar one.

**Figure 5 pharmaceuticals-18-00282-f005:**
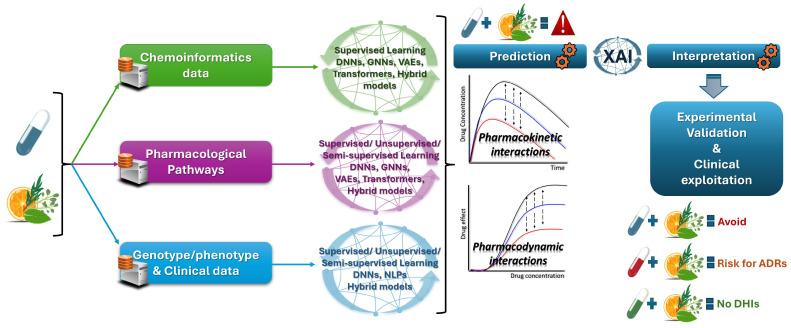
Matching AI tools to different data types (chemoinformatic data, pharmacological pathways, or clinical data) towards optimized DHI assessments.

**Table 1 pharmaceuticals-18-00282-t001:** Databases with data for building AI models towards DHI studies.

Database	Website	Description	Data Types Provided
DrugBank	https://www.drugbank.ca, accessed on 10 January 2025	Provides detailed information on drugs, targets, interactions, and pharmacokinetics for FDA-approved and experimental drugs.	Drug interactions, molecular structures, pharmacokinetics, pathways
TCMID (Traditional Chinese Medicine Integrated Database)	https://bidd.group/TCMID/, accessed on 10 February 2025	Integrates data on Traditional Chinese Medicine (TCM) herbs, chemical components, and related diseases.	Herb-ingredient interactions, targets, TCM herb data
STITCH (Search Tool for Interactions of Chemicals)	http://stitch.embl.de/, accessed on 10 January 2025	A comprehensive tool for exploring interactions between chemicals, drugs, and small molecules based on experimental and predicted data.	Chemical interaction networks, experimental and predicted data
TCMSP (Traditional Chinese Medicine Systems Pharmacology Database)	https://tcmsp-e.com/, accessed on 10 January 2025	Contains information on the pharmacokinetics and molecular properties of herbs in Traditional Chinese Medicine, including ADME profiles.	ADME properties, molecular targets, pharmacokinetics of herbs
PharmGKB	https://www.pharmgkb.org/, accessed on 10 January 2025	A pharmacogenomics knowledge resource that provides information on how drugs interact with genes and affect patient outcomes.	Pharmacogenomic interactions, genetic variants affecting drug responses
KEGG (Kyoto Encyclopedia of Genes and Genomes)	https://www.kegg.jp/, accessed on 10 January 2025	A database integrating biological pathways, drug actions, and metabolism, useful for studying the molecular basis of drug–herb interactions.	Pathway analysis, drug metabolism, gene and protein interactions
ChEMBL	https://www.ebi.ac.uk/chembl/, accessed on 10 January 2025	A large-scale bioactivity database that provides data on drug-like molecules and their biological activities for drug discovery and interactions research.	Bioactivity data, chemical structures, molecular targets
RxNorm	https://www.nlm.nih.gov/research/umls/rxnorm/, accessed on 10 January 2025	Provides normalized names and vocabularies for clinical drugs, facilitating standardized data integration for drug interaction studies.	Drug vocabularies, standardized drug names, interaction data
Therapeutic Target Database (TTD)	https://idrblab.net/ttd/, accessed on 10 January 2025	Provides information about therapeutic drug targets and corresponding drugs, relevant for assessing how herbs might influence therapeutic efficacy.	Therapeutic targets, drug-target interactions, clinical use data
PubChem	https://pubchem.ncbi.nlm.nih.gov/, accessed on 10 January 2025	One of the largest databases of chemical molecules and their activities. It includes information on chemical structures, properties, biological activities, and pharmacophores.	Chemical structures, bioactivity data, pharmacophore models, molecular interactions
BindingDB	https://www.bindingdb.org/, accessed on 10 January 2025	A database of measured binding affinities for drug-target interactions. Provides valuable data for pharmacophore modeling and structure-based drug design.	Drug-target interactions, binding affinities, molecular docking data
ZINC	https://zinc.docking.org/, accessed on 10 January 2025	A free database of commercially available compounds for virtual screening. It includes 3D models and is widely used for pharmacophore and drug interaction modeling.	3D chemical structures, pharmacophore data, virtual screening tools
SuperPred	https://prediction.charite.de/, accessed on 10 January 2025	A prediction tool for linking chemical structures to targets and diseases. It is useful for identifying possible interactions and repurposing opportunities.	Chemical–target interactions, disease–target associations, pharm
CMap	https://www.broadinstitute.org/connectivity-map-cmap, accessed on 10 January 2025	Library containing over 1.5 M gene expression profiles from ~5000 small-molecule compounds, and ~3000 genetic reagents, tested in multiple cell types. Includes functional relationships between drugs, genes and diseases.	Functional relationships between drugs, genes and diseases
ChemSpider	https://www.chemspider.com/, accessed on 10 January 2025	A free chemical structure database providing access to millions of compounds and their associated properties.	Chemical structures, molecular weights, melting points, boiling points, and other compound properties.
UniProt	https://www.uniprot.org/, accessed on 1 Februray 2025	A comprehensive protein sequence and functional information database, widely used in biological research.	Protein sequences, functional annotations, protein–protein interactions, subcellular locations, post-translational modifications.
Reactome	https://reactome.org/, accessed on 1 Februray 2025	A free, open-source database of biological pathways, providing detailed information about molecular processes	Biological pathways, gene sets, protein–protein interactions, signaling events, and metabolic pathways
PDBbind	https://www.pdbbind-plus.org.cn/, accessed on 1 Februray	A database of experimentally measured protein–ligand binding affinities, mainly used to evaluate and refine docking simulations and binding predictions	A database of experimentally measured protein–ligand binding affinities, mainly used to evaluate and refine docking simulations and binding predictions
IntAct	https://www.ebi.ac.uk/intact, accessed on 1 Februray	A database focused on molecular interaction data, providing an extensive collection of protein–protein interactions to aid in understanding cellular processes.	Protein–protein interaction data, interaction annotations, experimental methods, data on interaction networks.

## Data Availability

Data sharing is not applicable.

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
