# Peer review of "Artificial Intelligence Models and Tools for the Assessment of Drug–Herb Interactions"

_pharmaceuticals, 2025, doi:10.3390/ph18030282_

Round 1
Reviewer 1 Report
Comments and Suggestions for Authors
In this review the authors discuss about different AI approaches and how they can be used to provide valuable mechanistic insights for the prediction of interactions between natural herbs and drugs in detail. The review is clear and up to the point and can be considered for publication after revision.
1) The authors should discuss some example or relevant studies as well wherein AI approaches has been used for similar applications.
2) Figure 1 is not visible and it should be revised.
Author Response
In this review the authors discuss about different AI approaches and how they can be used to provide valuable mechanistic insights for the prediction of interactions between natural herbs and drugs in detail. The review is clear and up to the point and can be considered for publication after revision.
Thank you for your comments for our manuscript. Please see below our answers to the points that are raised.
1) The authors should discuss some example or relevant studies as well wherein AI approaches has been used for similar applications.
Answer: Thank you for your comment. AI is transforming drug research by improving efficiency across multiple stages. In drug design, AI-driven modeling accelerates candidate identification, while in chemical synthesis, it optimizes reaction pathways and uncovers novel synthesis routes. AI also facilitates drug repurposing by analyzing biomedical data to identify new therapeutic uses for existing drugs. Furthermore, high-throughput virtual screening and deep learning methods enhance the identification of promising drug candidates from vast chemical libraries with high accuracy. Notably, AI has significantly advanced various fields of pharmacology by uncovering multi-target interactions. In particular, deep learning (DL) models can integrate diverse data sources—including molecular structures, biological networks, and clinical data—making drug-drug interaction (DDI) predictions more scalable, accurate, and insightful for understanding adverse effects and synergistic potentials. We could provide a comprehensive analysis, but we believe it may cause the already lengthy review to drift from its primary focus. However, we have revised the introduction to include additional references and examples of systems that incorporate AI for DDI analysis.
See Lines 115-127 in our revised manuscript.
2) Figure 1 is not visible and it should be revised.
Answer: Please see the revised version of figure 1.
Reviewer 2 Report
Comments and Suggestions for Authors
Manuscript “AI Models and Tools for the Assessment of Drug-Herb Interactions” is a review in the fields of pharmaceutical chemistry and machine learning. It is a very promising area of science due to the huge opportunities given by the AI. Sciences which generated tons of raw data could benefit from AI-assisted processing of such data. Thus this manuscript describing the processing of data on interaction between drugs and herbs (food additives) is relevant.
Review consists of the following sections: “Exploitation of chemoinformatics, pharmacological pathways and clinical data databases for DHI prediction”; “AI methods for DHI prediction” with many subsections; Explainable AI (XAI) in drug-herb interactions; Mapping AI-Models to Data Types for assessment of DHIs; DHI and AI tools under the prism of network pharmacology, systems biology and computational toxicology along with introduction and conclusions.
Review is based on 172 literature references most of which are recent.
There are problems with section numbering in the manuscript.
3. AI methods for DHI prediction. Then goes subsections 3.1 / 3.1.1 / 3.1.2 / 3.1.3
And then 4.1 which probably should be 3.1.4 and the same with 4.1, 5.1, 6.1, 7.1
After that, section “8. Explainable AI (XAI) in drug-herb interactions” comes.
Please correct numbering between section 3 and section 8.
After that the manuscript could be accepted.
Author Response
Manuscript “AI Models and Tools for the Assessment of Drug-Herb Interactions” is a review in the fields of pharmaceutical chemistry and machine learning. It is a very promising area of science due to the huge opportunities given by the AI. Sciences which generated tons of raw data could benefit from AI-assisted processing of such data. Thus this manuscript describing the processing of data on interaction between drugs and herbs (food additives) is relevant.
Review consists of the following sections: “Exploitation of chemoinformatics, pharmacological pathways and clinical data databases for DHI prediction”; “AI methods for DHI prediction” with many subsections; Explainable AI (XAI) in drug-herb interactions; Mapping AI-Models to Data Types for assessment of DHIs; DHI and AI tools under the prism of network pharmacology, systems biology and computational toxicology along with introduction and conclusions.
Review is based on 172 literature references most of which are recent.
There are problems with section numbering in the manuscript.
- AI methods for DHI prediction. Then goes subsections 3.1 / 3.1.1 / 3.1.2 / 3.1.3
And then 4.1 which probably should be 3.1.4 and the same with 4.1, 5.1, 6.1, 7.1
After that, section “8. Explainable AI (XAI) in drug-herb interactions” comes.
Please correct numbering between section 3 and section 8.
After that the manuscript could be accepted.
Answer: Thank you for your comments on our work. We have updated the sections as you suggested so now the numbering is as follows:
- Introduction
- Exploitation of chemoinformatics, pharmacological pathways and clinical data databases for DHI prediction.
- AI methods for DHI prediction
3.1. Traditional Machine Learning Approaches
3.1.1. Supervised Learning
3.1.2. Unsupervised Learning/Dimensionality reduction
3.1.3. Semi-Supervised Learning
3.2. Deep Learning methods
3.2.1. Deep Neural Networks (DNNs)
3.2.2. Graph-based approaches
3.2.3. Natural Language Processing and Text Mining
3.2.4. Generative modelling-based approaches
3.3. Hybrid and integrative approaches
- Explainable AI (XAI) in drug-herb interactions
- Mapping AI-Models to Data Types for assessment of DHIs
- DHI and AI tools under the prism of network pharmacology, systems biology and computational toxicology
- Ethical and Practical Challenges of AI in DHI Prediction
- Conclusion
Reviewer 3 Report
Comments and Suggestions for Authors
The article “AI Models and Tools for the Assessment of Drug-Herb Interactions” by Spanakis et al. have reviewed the available AI tools employed for predicting Drug-Herb interactions at different levels. The article has been well written and comprehensively covers the related literature. The authors have explained a number of interaction tools and their theories to explain the drug-herb interaction. Some points need attention;
The utility of molecular docking to predict possible interactions is missing in literature.
Furthermore, the pharmacokinetic profile of a drug can be well understood by carrying out the ADMET studies which is not described. Authors may clarify the reason of non-inclusion of following databases are missing from the table i.e., Chemspider, BindingMOAD, Uniprot, Reactome, KEGG, PDBbind, Intact etc.
Additionally, there are some article and spacing errors in the manuscript which should be rechecked thoroughly.
The references need formatting corrections.
Recheck the caption of table 1.
Line 229: Remove extra full stop.
Line 404, 782: Remove extra space after full stop.
Line 293, 738, 754, 776: Replace “an” with “a” when used with herb.
Year is missing in ref. 5.
Year should be bold in ref. 12, 13, 18, 22, 25, 26, 28, 29 and others.
Author Response
The article “AI Models and Tools for the Assessment of Drug-Herb Interactions” by Spanakis et al. have reviewed the available AI tools employed for predicting Drug-Herb interactions at different levels. The article has been well written and comprehensively covers the related literature. The authors have explained a number of interaction tools and their theories to explain the drug-herb interaction. Some points need attention;
The utility of molecular docking to predict possible interactions is missing in literature.
Answer: Thank you for your comment. Our work focuses more on techniques that are related to AI. Molecular docking is not typically considered an AI approach but it can be part of the overall process; it is a computational method used in structural bioinformatics. For that reason, it is not thoroughly analyzed. However, we added a phrase within the text lines 186-189.
Molecular docking simulations can reveal the synergistic, additive, and multi-target (SAM) effects of herbal medicines by quantifying network interactions, estimating kinetic parameters, and enhancing screening efficiency when combined with pharmacophore modeling[63].
Furthermore, the pharmacokinetic profile of a drug can be well understood by carrying out the ADMET studies which is not described. Authors may clarify the reason of non-inclusion of following databases are missing from the table i.e., Chemspider, BindingMOAD, Uniprot, Reactome, KEGG, PDBbind, Intact etc.
Answer: Thank you for your comment. DHIs are often related with ADMET and PK-related mechanisms. We could go on details for ADMET studies, but the manuscript may drift from its purpose to present AI tools and models for DHIs. But we make a small reference in section 5. Moreover, we added some of the missing databases to table 1. We hope it suffices.
Additionally, there are some article and spacing errors in the manuscript which should be rechecked thoroughly.
- The references need formatting corrections.
- Recheck the caption of table 1.
- Line 229: Remove extra full stop.
- Line 404, 782: Remove extra space after full stop.
- Line 293, 738, 754, 776: Replace “an” with “a” when used with herb.
- Year is missing in ref. 5.
- Year should be bold in ref. 12, 13, 18, 22, 25, 26, 28, 29 and others.
Answer: Thank you for your comment, and we appreciate your effort in identifying these issues. We edited all the points made above. We used the Mendeley tool for references, and any mistakes may have resulted from Mendeley’s automated export/ import process. As the Mendeley Cite software states for Pharmaceuticals’ citation settings "This citation style language is predefined and cannot be changed". We have tried to address the proposed corrections, along with others we identified, and we hope the issues are now resolved or it will be further during proofing of the manuscript.
Reviewer 4 Report
Comments and Suggestions for Authors
Title: AI Models and Tools for the Assessment of Drug-Herb Interactions
1. How does your paper contribute to the existing body of knowledge in AI applications for DHI studies? Do you believe it offers any particularly novel insights, frameworks, or approaches that set it apart from prior work?
2. Could you elaborate on how your paper demonstrates the potential of XAI in this field? Are there specific examples or case studies showing how XAI can enhance understanding or aid clinical decision-making?
3. One challenge in this field is translating computational predictions into experimental or clinical validation. How does your work address this gap? Are there examples or discussions in the paper to show how these predictions can be tested and validated in real-world settings?
4. The solutions you’ve proposed to tackle the identified challenges are intriguing, but are they practical? Could you provide more evidence or examples to show how these solutions could be realistically implemented?
5. Integrating chemoinformatic, pharmacological, and clinical data is critical for robust AI models. How thoroughly does your paper address this integration? Have you explored both the potential benefits and the challenges involved in combining these datasets?
6. The use of technical jargon (e.g., "latent representations," "transformer-based models") may limit accessibility for non-expert readers; therefore, including a glossary or sidebars to explain complex terms in layman's terms would enhance clarity and understanding.
7. Although the manuscript mentions Explainable AI, it fails to elaborate on its potential to enhance the interpretability and clinical utility of AI models in the context of DHIs; thus, incorporating specific examples or proposed frameworks for integrating XAI techniques would better demonstrate their relevance to clinical or experimental contexts.
8. The manuscript makes numerous claims about AI's potential benefits, such as improving clinical outcomes and enabling precision medicine, without providing sufficient citations or supporting evidence. Furthermore, it overlooks critical discussions on limitations and risks, including biases in AI models and ethical considerations when integrating patient-specific data. Strengthen the manuscript by including robust references to substantiate key claims and addressing the ethical and practical limitations associated with AI applications in DHIs.
Recommendation: Consider to acceptance after Major Revision.
Comments on the Quality of English Language1. Some sentences are unnecessarily complex and can be simplified for better readability. For example, "The variability and inconsistency in herbal product data are acknowledged, but the manuscript does not propose practical solutions or methodologies to overcome these challenges" could be revised to "While the variability in herbal product data is acknowledged, the manuscript lacks practical solutions to address these challenges."
2. The manuscript uses technical terms like latent representations and transformer-based models without explanation, which may confuse non-specialist readers. Adding brief definitions or a glossary would enhance accessibility.
3. Grammar and Syntax: A few sentences have grammatical issues or awkward phrasing. For instance, "These models, including hybrid approaches like transfer learning, are well-equipped to interpret chemical structures and predict DDIs/DHIs based on molecular properties" could be revised to "These models, including hybrid approaches like transfer learning, effectively interpret chemical structures and predict DDIs/DHIs from molecular properties."
4. Consistency: The manuscript occasionally shifts between terms such as drug-herb interactions and herb-drug interactions. Consistent terminology throughout the text would improve coherence.
Author Response
- How does your paper contribute to the existing body of knowledge in AI applications for DHI studies? Do you believe it offers any particularly novel insights, frameworks, or approaches that set it apart from prior work?
Answer: This review article aims to contribute to the existing body of knowledge in AI applications for drug-herb interaction (DHI) studies by drawing critical parallels with drug-drug interaction (DDI) prediction methods while highlighting the unique challenges posed by DHIs. Our work provides a comprehensive overview of AI techniques that have been successfully applied in related contexts, including deep learning, knowledge graphs, and network-based approaches, and discussing their potential adaptation to DHI prediction. A key contribution of this review is its emphasis on how advanced AI methodologies—such as transfer learning, semi-supervised learning, multi-modal AI, and generative models—can help overcome the challenges of limited labeled data, complex multi-component interactions, and variability in herbal composition. By synthesizing existing research and proposing concrete pathways for future AI-driven advancements in DHI prediction, this review serves as both a foundational reference and a forward-looking perspective, setting the stage for more accurate, interpretable, and clinically relevant AI applications in herbal medicine safety and precision medicine.
- Could you elaborate on how your paper demonstrates the potential of XAI in this field? Are there specific examples or case studies showing how XAI can enhance understanding or aid clinical decision-making?
Answer: Explainable AI (XAI) is gaining attention in drug research, as it addresses the need for transparency when complex AI models are exploited. In the case o drug interactions, either DDIs or DHIs this is particularly important not only to determine that two compounds may interact but also to understand the underlying mechanisms and pathways involved. By providing mechanistic insights, XAI can help us decipher whether an interaction is due to metabolic interference, receptor binding competition, or synergistic effects, ultimately leading to more informed decision-making and targeted experimental validation. We have included the above explanation on the potential and usefulness of XAI and also incorporated specific examples based on the reference below, which is a systematic review on applications of XAI for DDIs, in section 4 (see also lines 657-683 for examples and reference 45)
- One challenge in this field is translating computational predictions into experimental or clinical validation. How does your work address this gap? Are there examples or discussions in the paper to show how these predictions can be tested and validated in real-world settings?
Answer: Translating computational predictions of DHIs into experimental and clinical validation is a critical step for ensuring their reliability and clinical relevance. Computationally predicted DHIs can be validated through a combination of in vitro assays, in vivo animal models, retrospective clinical data analysis, prospective clinical studies, and multi-omics approaches. see our addition:
"Experimental methods that could complement AI predictions range from in vitro, mi-crosomal, and cell culture assays to in vivo animal experiments, physiologically based models, and even clinical trials. These approaches could consider AI driven results for single-ingredient herbs, crude extracts or standardized herbal formulas, collectively building evidence-based data on the potential clinical significance of DHIs for clinical utilization [154,155,157,174–176]."
We have added a new section in the manuscript on the ethical and practical use of AI in DHI prediction, where we also discuss the importance of validating computational predictions in real-world settings.
- The solutions you’ve proposed to tackle the identified challenges are intriguing, but are they practical? Could you provide more evidence or examples to show how these solutions could be realistically implemented?
Answer: We would like to thank the reviewer for this comment. We have added a new section in the manuscript on the ethical and practical use of AI in DHI prediction. Details about this section are included in this document, following all of our responses to the reviewers.
- Integrating chemoinformatic, pharmacological, and clinical data is critical for robust AI models. How thoroughly does your paper address this integration? Have you explored both the potential benefits and the challenges involved in combining these datasets?
Answer: Thank you for this comment. Indeed, integrating heterogeneous data is a very important step in improving AI model performance. In this work, we discuss the advantages and challenges of combining diverse datasets and we briefly mention popular solutions to deal with data integration, in Sections 2 and 5.
- The use of technical jargon (e.g., "latent representations," "transformer-based models") may limit accessibility for non-expert readers; therefore, including a glossary or sidebars to explain complex terms in layman's terms would enhance clarity and understanding.
Answer: Thank you for this suggestion. We have included a glossary as an Appendix to enhance clarity, readability and accessibility. More information about this glossary is included in this document, following all of our responses to the reviewers.
- Although the manuscript mentions Explainable AI, it fails to elaborate on its potential to enhance the interpretability and clinical utility of AI models in the context of DHIs; thus, incorporating specific examples or proposed frameworks for integrating XAI techniques would better demonstrate their relevance to clinical or experimental contexts.
Answer: This comment is similar with comment 2 so we provide a similar answer. XAI is particularly important not only to determine that two compounds may interact but also to understand the underlying mechanisms and pathways involved. By providing mechanistic insights, XAI can help clinicians and researchers decipher whether an interaction is due to metabolic interference, receptor binding competition, or synergistic effects, ultimately leading to more informed decision-making and targeted experimental validation.
We have included the above explanation on the potential and usefulness of XAI and also incorporated specific examples based on the reference below, in section 4.
https://doi.org/10.1016/j.csbj.2022.04.021
- The manuscript makes numerous claims about AI's potential benefits, such as improving clinical outcomes and enabling precision medicine, without providing sufficient citations or supporting evidence. Furthermore, it overlooks critical discussions on limitations and risks, including biases in AI models and ethical considerations when integrating patient-specific data. Strengthen the manuscript by including robust references to substantiate key claims and addressing the ethical and practical limitations associated with AI applications in DHIs.
Answer: We have added a new section in the manuscript on the ethical and practical use of AI in DHI prediction and we believe it addresses the comments raised.
- Some sentences are unnecessarily complex and can be simplified for better readability. For example, "The variability and inconsistency in herbal product data are acknowledged, but the manuscript does not propose practical solutions or methodologies to overcome these challenges" could be revised to "While the variability in herbal product data is acknowledged, the manuscript lacks practical solutions to address these challenges."
Answer: This phrase is not included in our initial text so we cannot revise it. But we have revised the syntax and phrasing in our manuscript to ensure improved readability and comprehensibility.
- The manuscript uses technical terms like latent representations and transformer-based models without explanation, which may confuse non-specialist readers. Adding brief definitions or a glossary would enhance accessibility.
Answer: We added a glossary term as an appendix with brief definitions
- Grammar and Syntax: A few sentences have grammatical issues or awkward phrasing. For instance, "These models, including hybrid approaches like transfer learning, are well-equipped to interpret chemical structures and predict DDIs/DHIs based on molecular properties" could be revised to "These models, including hybrid approaches like transfer learning, effectively interpret chemical structures and predict DDIs/DHIs from molecular properties."
Answer: We believe our phrasing is more scientifically sound. Without sufficient data and evidence, we cannot state with certainty that transfer learning models ‘effectively’ interpret. However, they are designed for this purpose and will be evaluated based on their performance.
- Consistency: The manuscript occasionally shifts between terms such as drug-herb interactions and herb-drug interactions. Consistent terminology throughout the text would improve coherence.
Answer: We consistently use drug-herb interaction and the acronym DHI in the abstract, main body, tables and figurers of our manuscript. However, the term ‘herb-drug interactions’ appears in the Refencences (only), as the authors of the cited work have used that term instead. We cannot change that.
Round 2
Reviewer 3 Report
Comments and Suggestions for Authors
The authors have now revised the manuscript and can now be accepted for publication in current form.
Author Response
The authors have now revised the manuscript and can now be accepted for publication in current form.
Answer: Thank you very much for endorsing our publication
Reviewer 4 Report
Comments and Suggestions for Authors
None
Author Response
None
Answer: We are happy to hear that we succesfully address all the comments